# DISTRIBUTED SKELLAM MECHANISM: A NOVEL APPROACH TO FEDERATED LEARNING WITH DIFFERENTIAL PRIVACY

## ABSTRACT

Deep neural networks have strong capabilities of memorizing the underlying training data; on the flip side, unintended data memorization can be a serious privacy concern. An effective and rigorous approach to addressing this problem is to train models with *differential privacy* (*DP*), which provides information-theoretic privacy guarantees by injecting random noise to the gradients. This paper focuses on the scenario where sensitive data are distributed among individual participants, who jointly train a model through *federated learning*, using both *secure multiparty computation* (*MPC*) to ensure the confidentiality of individual gradients, and differential privacy to avoid data leakage in the resulting model. A major challenge in this setting is that common mechanisms for enforcing DP in deep learning, which inject *real-valued noise*, are fundamentally incompatible with MPC, which exchanges *finite-field integers* among the participants. Consequently, most existing DP mechanisms require rather high noise levels, leading to poor model utility.

Motivated by this, we design and develop *distributed Skellam mechanism* (DSM), a novel solution for enforcing differential privacy on models built through an MPC-based federated learning process. Compared to existing approaches, DSM has the advantage that its privacy guarantee is independent of the dimensionality of the gradients; further, DSM allows tight privacy accounting due to the nice composition and sub-sampling properties of the Skellam distribution, which are key to accurate deep learning with DP. The theoretical analysis of DSM is highly non-trivial, especially considering (i) the complicated math of differentially private deep learning in general and (ii) the fact that the Skellam distribution is rather complex, and to our knowledge, has not been applied to an iterative and sampling-based process, i.e., stochastic gradient descent. Meanwhile, through extensive experiments on various practical settings, we demonstrate that DSM consistently outperforms existing solutions in terms of model utility by a large margin.

## 1 INTRODUCTION

Deep neural networks, especially large-scale ones such as GPT-3 (Brown et al., 2020), are known for their excellent memorization capabilities (Song et al., 2017; Feldman, 2020; Zhang et al., 2021). However, it is rather difficult to control what exactly the neural net memorizes, and unintended data memorization can be a serious concern when the underlying training data contains sensitive information (Carlini et al., 2019). For instance, consider a bank that trains a GPT-like language model on call center transcripts. Due to data memorization, it is possible to extract sensitive information by letting the model auto-complete a prefix, e.g., "my account number is: ＿". Clearly, if such a model (or its API) is ever exposed to the adversary, it becomes a ligation machine as attackers can attempt with various prefixes to extract sensitive data, and subsequently sue the bank for privacy violations. Shokri et al. (2017) report that simple and intuitive measures often fail to provide sufficient protection, and the only way found to completely address the issue is to train the model with the rigorous guarantees of *differential privacy* (*DP*) (Dwork et al., 2006).

This paper focuses on the scenario that multiple individual participants jointly train a machine learning model using *federated learning* (*FL*) (McMahan et al., 2017) through distributed stochastic gradient descent (SGD) (McDonald et al., 2010; Dean et al., 2012; Coates et al., 2013; Abadi et al.,

2016a). Specifically, in every iteration, each individual computes the gradients with respect to the current model weights based on her own data; then, gradients from all participants are aggregated to update the model. Note that the gradients from each individual may reveal sensitive information about her private dataset (Shokri et al., 2017; Pyrgelis et al., 2018; Yeom et al., 2018; Nasr et al., 2019; Melis et al., 2019). A common approach to addressing this problem is by employing a *secure multiparty computation* (*MPC*) protocol (Yao, 1986; Chaum et al., 1987; Gennaro et al., 2002; Ishai et al., 2010; Beimel et al., 2014; Cramer et al., 2015; Ananth et al., 2018), which computes the aggregate gradients while preserving the confidentiality of the gradients from each individual participant. One advantage of MPC is that it is a decentralized approach that does not require a trusted third party, which can be difficult to establish in some applications, e.g., in finance and healthcare.

Note that although MPC protects individuals' privacy in the gradient update process by concealing the gradient values of each participant, it does not provide any protection against data extraction attacks caused by unintended data memorization (Dwork et al., 2015; Song & Shmatikov, 2019; Melis et al., 2019; Song & Shmatikov, 2020). As mentioned earlier, an effective methodology to defend against such attacks is to perturb the gradients to satisfy differential privacy (Shokri et al., 2017). Since there is no trusted third-party in our setting, such gradient perturbations need to done in a decentralized fashion, i.e., each FL participant adds noise to her own gradients, such that the aggregated gradients over all participants satisfies DP, which is referred to as distributed differential privacy (Goryczka et al., 2013; Kairouz et al., 2021).

Although gradient perturbation under DP has been studied in previous work (notably, DPSGD (Abadi et al., 2016b)), it is far from trivial to adapt centralized DP solutions to our setting, due to a fundamental problem: that the MPC protocol requires gradients to be represented as *integers* (more precisely, finite field elements (Paillier, 1999; Bonawitz et al., 2017; Bell et al., 2020)). DPSGD, on the other hand, injects *real-valued* Gaussian noise to the gradients. Although real numbers can be quantized and (approximately) represented using large integers, the quantized random noise have rather different mathematical properties, which render a tight privacy cost analysis much more difficult, especially under the decentralized setting of FL. For instance, a nice property of the continuous Gaussian distribution is that summing up $n$ continuous noise values following i.i.d. unit-variance Gaussian distribution results in an amplified continuous Gaussian noise of variance $n$. This property does not hold, however, if the Gaussian noise values are first quantized before aggregated. Further, the privacy analysis (specifically, the moment accountant analysis technique) of the DPSGD algorithm also replies on other important properties of the continuous Gaussian distribution, which do not hold when the noise is quantized. Hence, DPSGD does not directly apply to our setting. This issue has been neglected by many existing distributed DP solutions, e.g., (Goryczka et al., 2013; Valovich & Aldà, 2017; Truex et al., 2019).

**Existing Solutions.** Agarwal et al. (2018) propose cpSGD, which injects binomial noise (i.e., the sum of multiple binary values drawn from independent Bernoulli trials) to the discretized gradients at each participant of FL, to satisfy DP. Similar to Gaussian noise in the continuous domain, binomial noise can also be aggregated, i.e., the sum of multiple i.i.d. binomial noise values also follows a binomial distribution. However, compared to the continuous Gaussian distribution, existing theoretical tools for analyzing binomial noise aggregation leads to rather loose bounds; further, the bionomial distribution is incompatible with the moment accountant analysis technique in DPSGD (Kairouz et al., 2021). Consequently, cpSGD leads to poor utility, as demonstrated in Section 4.

Recently, the *distributed discrete Gaussian mechanism* (DDG) (Kairouz et al., 2021) addresses the above issues by injecting independent *discrete Gaussian noise* (Canonne et al., 2020) to the gradients at each participant. Similar to the binomial distribution, the discrete Gaussian distribution is also defined over an integer domain; meanwhile, DDG is fully compatible with the moment accountant analysis technique in DP-SGD, and, thus, enjoys the tight privacy cost analysis. However, the discrete Gaussian distribution is not aggregatable, meaning that the sum of noise drawn from multiple i.i.d. discrete Gaussian distributions does not follow another discrete Guassian distribution, which renders analysis difficult under in the decentralized setting of FL, and leads to looser bounds in the privacy analysis. Further, the privacy guarantee of the aggregated noise in DDG degrades linearly with the dimensionality $d$ of the gradients, leading to poor scalablity to large neural networks.

**Our contribution.** In this work, we propose a new mechanism for enforcing distributed differential privacy for federated learning: the *distributed Skellam mechanism* (DSM), which injects random noise drawn from the symmetric Skellam distribution. Although the Skellam distribution has been

used before in the DP literature (Valovich & Aldà, 2017), the privacy analysis therein does not cover the decentralized setting of FL, or the iterative, sampling-based SGD algorithm, both of which require highly non-trivial mathematical analysis.

Specifically, we prove that DSM satisfies both Rényi-DP and $(\epsilon, \delta)$-DP, defined in Section 2. Similar to our competitor DDG described above, DSM is compatible with the DPSGD framework and its moment accountant analysis technique, leading to tight bounds on the privacy loss analysis. Meanwhile, unlike DDG, the privacy guarantees of DSM are independent of the dimensionality of the gradients, which scales well to large models. Further, similar to the continuous Gaussian distribution (and unlike the discrete Gaussian distribution in DDG), i.i.d. Skellam noise values can be aggregated to form an amplified noise that still follows the Skellam distribution, which leads to clean and elegant proofs in the decentralized setting of FL, and tight bounds in the privacy analysis.

We apply DSM to federated learning with distributed SGD, with quantized gradients, e.g., as required by the MPC protocol, and present the complete training algorithm. Extensive experiments using benchmark datasets show that our solution leads to consistent and significant utility gains over its competitors, under a variety of settings with different privacy and communication constraints.

## 2 PRELIMINARIES

A random variable $Y$ follows a Poisson distribution of parameter $\lambda$ if its probability distribution is $\Pr[Y = k] = \frac{\exp(-\lambda)\lambda^k}{k!}, k = 0, 1, 2, \ldots$. Both the mean and variance of $Y$ is $\lambda$. A random variable $Z$ follows a Skellam distribution if it is the difference between two independent Poisson variables $Y_1$ and $Y_2$. In this work, we restrict our attention to the case where $Y_1$ and $Y_2$ have the same parameter $\lambda$. In that case, the probability distribution of $Z$ is

$$\Pr[Z = k] = \exp(-2\lambda)I_{|k|}(2\lambda), k = 0, \pm 1, \pm 2, \ldots, \tag{1}$$

where $I_v(u) \triangleq \sum_{h=0}^{\infty} \frac{1}{h!\Gamma(h+v+1)} \left(\frac{u}{2}\right)^{2h+v}$ is the modified Bessel function of the first kind. We write that $Z \sim \mathrm{Sk}(\lambda, \lambda)$. By linearity of expectation, $Z$ has mean 0 and variance $2\lambda$.

We say that two datasets $X$ and $X'$ are neighboring if one can be obtained by adding or removing one tuple from the other. The main idea of differential privacy (DP) is to ensure that the outcomes of a randomized mechanism on neighboring datasets are always similar; intuitively, this provides plausible deniability on whether a given data record $x$ belongs to the dataset $X$ or not, and, thus, protects the privacy of the individual whose record is $x$. A classic definition of differential privacy is $(\epsilon, \delta)$-DP (Dwork et al., 2006), as follows.

**Definition 1** ($(\epsilon, \delta)$-Differential Privacy (Dwork et al., 2006))**.** *A randomized mechanism $\mathcal{M}$ satisfies $(\epsilon, \delta)$-differential privacy (DP) if*

$$\Pr[\mathcal{M}(X) \in \mathcal{O}] \leq \exp(\epsilon) \cdot \Pr[\mathcal{M}(X') \in \mathcal{O}] + \delta, \tag{2}$$

*for any set of output $\mathcal{O} \subseteq Range(\mathcal{M})$ and any neighboring datasets $X$ and $X'$.*

Note that $(\epsilon, \delta)$-DP can be considered as a worst-case privacy guarantee for a mechanism, as it enforces an upper bound on the probability ratio of all possible outcomes. An alternative definition called Rényi-Differential Privacy (RDP) (Mironov, 2017), which is built upon the concept of Rényi Divergence, considers the average case privacy guarantee instead.

**Definition 2** (Rényi Divergence (van Erven & Harremoës, 2014))**.** *Assuming that distributions $P$ and $Q$ are defined over the same domain, and $P$ is absolute continuous with respect to $Q$, then the Rényi divergence of $P$ from $Q$ of finite order $\alpha \in (0, 1) \cup (1, \infty)$ is defined as:*

$$D_\alpha(P\|Q) = \frac{1}{\alpha - 1} \log \mathbb{E}_{X \sim P} \left[ \left(\frac{P(X)}{Q(X)}\right)^{\alpha - 1} \right], \tag{3}$$

*where we adopt the convention that $\frac{0}{0} = 0$ and $\frac{y}{0} = \infty$ for any $y > 0$, and the logarithm is with base $e$.*

**Definition 3** (Rényi Differential Privacy (Mironov, 2017))**.** *A randomized mechanism $\mathcal{M}$ satisfies $(\alpha, \tau)$-Rényi differential privacy (RDP) if $D_\alpha(\mathcal{M}(X)\|\mathcal{M}(X')) \leq \tau$ for all neighboring datasets $X$ and $X'$.*

Given a function of interest, the canonical way make it differentially private is to perturb its outcome through noise injection. Roughly speaking, the scale of the noise should be calibrated to the sensitivity of the function of interest (Dwork et al., 2006), formally defined as follows.

**Definition 4** (Sensitivity). *The sensitivity $S(F)$ of a function $F : \mathcal{D} \to \mathbb{R}^d$, denoted as $S(F)$, is defined as*

$$S(F) = \max_{X \sim X'} \|F(X) - F(X')\|,$$

*where $X \sim X'$ denotes that $X$ and $X'$ are neighboring datasets, and $\|\cdot\|$ is a norm.*

In particular, injecting continuous Gaussian noise sampled from $\mathcal{N}(0, \sigma^2)$ to each dimension of function $F$ satisfies $(\alpha, \frac{\alpha S^2(F)}{2\sigma^2})$-RDP (Mironov, 2017), where $S(F)$ stands for the $\mathcal{L}_2$ sensitivity of function $F$. In many applications, we also need to analyze the overall privacy guarantee of a mechanism consisting of multiple components (e.g., training neural networks with SGD). We have the following composition and sub-sampling lemmata for RDP mechanisms.

**Lemma 1** (Composition Lemma (Mironov, 2017)). *If mechanisms $\mathcal{M}_1, \ldots, \mathcal{M}_T$ satisfies $(\alpha, \tau_1), \ldots, (\alpha, \tau_T)$-RDP, respectively, then $\mathcal{M}_1 \circ \ldots \circ \mathcal{M}_T$ satisfies $(\alpha, \sum_{t=1}^T \tau_i)$-RDP.*

**Lemma 2** (Subsampling Lemma (Zhu & Wang, 2019; Mironov et al., 2019)). *Let $\mathcal{M}$ be a mechanism that satisfies $(l, \tau(l))$-RDP for $l = 2, \ldots, \alpha$ ($\alpha \in \mathbb{Z}, \alpha > 2$), and $S_q$ be a procedure that uniformly sample each record of the input data with probability $q$. Then $\mathcal{M} \circ S_q$ satisfies $(\alpha, \tau)$-RDP with*

$$\tau = \frac{1}{\alpha - 1} \log \left( (1-q)^{\alpha-1}(\alpha q - q - 1) + \sum_{l=2}^{\alpha} \binom{\alpha}{l}(1-q)^{\alpha-l}q^l e^{(l-1)\tau(l)} \right).$$

Finally, any mechanism that satisfies $(\alpha, \tau)$-RDP also satisfies $(\epsilon, \delta)$-DP, for values of $\epsilon$ and $\delta$ as follows.

**Lemma 3** (Converting $(\alpha, \tau)$-RDP to $(\epsilon, \delta)$-DP (Canonne et al., 2020)). *For any $\alpha \in (1, \infty)$, if $D_\alpha(\mathcal{M}(X)\|\mathcal{M}(X')) \leq \tau$ for any neighboring databases $X$ and $X'$, then $\mathcal{M}$ satisfies $(\epsilon, \delta)$-DP for*

$$\epsilon = \tau + \frac{\log(1/\delta) + (\alpha - 1)\log(1 - 1/\alpha) - \log(\alpha)}{\alpha - 1}. \tag{4}$$

## 3 DISTRIBUTED SKELLAM MECHANISM

Section 3.1 first formalizes the problem of distributed sum estimation, and presents the proposed *distributed Skellam mechanism* (DSM) for this problem. Section 3.2 establishes the privacy guarantees of DSM. Then, in Section 3.3, we apply DSM to our main problem setting: differentially private federated learning.

### 3.1 FORMAL DESCRIPTION

Suppose that an integer-valued multi-dimensional dataset $X = (x_1, \ldots, x_n)$ is distributed to $n$ individuals (referred to as *clients* in the following), where client $i$ possesses data point $x_i \in \mathbb{Z}^d$, for $i = 1, \ldots, n$. Without loss of generality, we assume that the $\mathcal{L}_2$ and $\mathcal{L}_\infty$ norms of each $x_i$ is bounded by constants $\Delta_2$ and $\Delta_\infty$, respectively, i.e., $\|x_i\|_2 \leq \Delta_2$ and $\|x_i\|_\infty \leq \Delta_\infty$ for any $x_i \in X$. An untrusted *server* aims to compute the (approximate) sum of the dataset, i.e., $\bar{x} = \sum_{i=1}^n x_i$, from the clients. The computation of $\bar{x}$ must satisfy differential privacy. In particular, we focus on the RDP definition (Definition 3), which can be converted to the classic $(\epsilon, \delta)$-DP (Definition 1) through Lemma 3.

Specifically, we aim to design a private mechanism $\mathcal{M}$, such that for all neighboring datasets $X, X'$,

$$D_\alpha(\mathcal{M}(X)\|\mathcal{M}(X')) \leq \tau, \tag{5}$$

for some $\alpha > 1$. We measure the error of $\mathcal{M}$ by

$$Err_{\mathcal{M}} = \max_{X \subset \mathbb{Z}^d} \mathbb{E} \left\| \mathcal{M}(X) - \sum_{x \in X} x \right\|_2^2, \tag{6}$$

---

**Algorithm 1:** Distributed Skellam mechanism

---

**Input:** Private dataset $X = (x_1, \ldots, x_n)$. Noise parameter $\lambda$. Secure aggregation protocol $\mathcal{A}(\cdot)$.
**Output:** $\tilde{x}$, a private estimation of $\sum_{x \in X} x$.

1 **for** $x_i \in X$ **do**
2     $\tilde{x}_i \longleftarrow x_i + Z$, where $Z = (Z^1, \ldots, Z^d)$ is sampled as in Eq. (7).
3 $\tilde{x} \longleftarrow \mathcal{A}(\tilde{x}_1, \ldots, \tilde{x}_n)$; secure aggregation.

---

where the expectation is taken over the randomness in $\mathcal{M}$.

Next, we present the proposed solution DSM for the above distributed sum estimation problem, outlined in Algorithm 1. Each client $i$ first independently samples a noise vector following the Skellam distribution, and injects this noise to her data (Line 2 in Algorithm 1). In particular, the client samples a random $d$-dimensional noise vector $Z = (Z^1, \ldots, Z^d)$, where each element is identically and independently distributed, and follows the Skellam distribution of parameter $\lambda$, i.e.,

$$Z^1, \ldots, Z^d \overset{i.i.d}{\sim} \text{Sk}(\lambda, \lambda). \tag{7}$$

We assume the clients have black-box access to a secure aggregation protocol $\mathcal{A}$, following the convention in (Agarwal et al., 2018; Kairouz et al., 2021). The clients then compute the sum of their perturbed data using $\mathcal{A}$, and release the sum to the server (Line 3 in Algorithm 1).

**Modular Arithmetic with communication constraints.** When each element $x_i$ in $X$ is represented using a limited number $B$ of bits, each client needs to perform an extra step of modulo operation depending on $B$ after injecting Skellam noise (Line 2), so as to avoid overflows. Accordingly, after obtaining the sum (Line 3), the server also needs to unwrap the modulo operation to recover the original noisy sum. We refer the reader to (Kairouz et al., 2021) for a detailed discussion on this issue. In our experiments in Section 4, we demonstrate that the proposed solution DSM achieves significant accuracy gains compared to existing approaches with various settings of the bit limit $B$.

## 3.2 THEORETICAL ANALYSIS

The following theorem states the privacy guarantee for Algorithm 1, which is the main theoretical result of this paper.

**Theorem 1** (Privacy guarantee of the distributed Skellam mechanism). *When* $1 < \alpha < \frac{2n\lambda}{\Delta_\infty} + 1$, *Algorithm 1 with noise parameter* $\lambda$ *satisfies* $(\alpha, \tau)$-*Rényi Differential Privacy with*

$$\tau = \frac{1.09\alpha + 0.91}{2} \cdot \frac{\Delta_2^2}{2n\lambda}. \tag{8}$$

*Proof.* To prove the privacy guarantee of Algorithm 1, it suffices to derive the privacy guarantee of the sum of $\{Sk_\lambda(x_i)\}_{i=1}^n$. Note that the secure aggregation protocol $\mathcal{A}$ (used as a black-box) ensures that only the sum of the perturbed data is revealed to the server. In addition, according to properties of the Skellam distribution, aggregating $n$ independent Skellam noise vectors $Z \sim Sk(\lambda, \lambda)$ results in a single Skellam noise $\bar{Z} \sim Sk(n\lambda, n\lambda)$ (Skellam, 1946). Hence, we only need to focus on the privacy guarantee achieved by $\bar{Z}$. Further, in what follows, we focus on the one dimensional case of the problem, i.e., $d = 1$. The proof in the general case where $d > 1$ then follows from the additivity of Rényi divergence for product distributions (van Erven & Harremoës, 2014).

Without loss of generality, we let $s > 0$, $Z \sim \text{Sk}(\lambda, \lambda)$ and compute $\Phi = \exp((\alpha-1)D_\alpha(Z+s\|Z))$. We first present several useful inequalities, whose proofs can be found in the appendix. Let

$$Q_{s,t}(v) := \prod_{i=1}^{s} \frac{v + i + \sqrt{(v+i)^2 + t^2}}{t}. \tag{9}$$

According to Lemma 4 in the appendix, we have:

$$\frac{I_{|v|}(t)}{I_{|v+s|}(t)} \leq Q_{s,t}(v). \tag{10}$$

In addition, for any $0 < a \le w$, according to Lemmata 5-7 in the appendix, we have

$$\frac{w + \sqrt{w^2 + 1}}{(a - w) + \sqrt{(a - w)^2 + 1}} \le e^{2w - a}, \tag{11}$$

$$\text{and } \left( w + \sqrt{w^2 + 1} \right) \cdot \left( (a - w) + \sqrt{(a - w)^2 + 1} \right) \le e^a. \tag{12}$$

By the definition of Skellam distributions and Eq. (9)

$$\Phi = \sum_{z=-\infty}^{\infty} \left( \frac{I_{|z|}(2\lambda)}{I_{|z+s|}(2\lambda)} \right)^{\alpha - 1} \Pr[Z = z] \quad \le \sum_{z=-\infty}^{\infty} (Q_{s,2\lambda}(z))^{\alpha - 1} \Pr[Z = z]$$

$$= \sum_{|z| \le s} (Q_{s,2\lambda}(z))^{\alpha - 1} \Pr[Z = z] + \sum_{z > s} \left( (Q_{s,2\lambda}(z))^{\alpha - 1} + (Q_{s,2\lambda}(-z))^{\alpha - 1} \right) \Pr[Z = z].$$

We compute

$$(Q_{s,2\lambda}(z))^{\alpha - 1} + (Q_{s,2\lambda}(-z))^{\alpha - 1} = (Q_{s,2\lambda}(z) Q_{s,2\lambda}(-z))^{\frac{\alpha - 1}{2}} \cdot \left( R_{s,2\lambda}(z) + \frac{1}{R_{s,2\lambda}(z)} \right),$$

where

$$R_{s,2\lambda}(z) := \left( \frac{Q_{s,2\lambda}(z)}{Q_{s,2\lambda}(-z)} \right)^{\frac{\alpha - 1}{2}}.$$

By Eq. (11), when $z \ge i$,

$$\frac{z + i + \sqrt{(z+i)^2 + (2\lambda)^2}}{-z + i + \sqrt{(-z+i)^2 + (2\lambda)^2}} = \frac{\frac{z+i}{2\lambda} + \sqrt{(\frac{z+i}{2\lambda})^2 + 1}}{\frac{-z+i}{2\lambda} + \sqrt{(\frac{-z+i}{2\lambda})^2 + 1}} < \exp\left(\frac{z}{\lambda}\right).$$

Therefore, when $z \ge s$, we have $\frac{Q_{s,2\lambda}(z)}{Q_{s,2\lambda}(-z)} < \exp\left(\frac{sz}{\lambda}\right)$, and $R_{s,2\lambda}(z) < \exp\left(\frac{s(\alpha-1)z}{2\lambda}\right)$. Since the function $v \mapsto v + \frac{1}{v}$ is increasing,

$$R_{s,2\lambda}(z) + \frac{1}{R_{s,2\lambda}(z)} < \exp\left(\frac{s(\alpha-1)z}{2\lambda}\right) + \exp\left(-\frac{s(\alpha-1)z}{2\lambda}\right).$$

By Eq. (12), when $z \ge i$,

$$\frac{z + i + \sqrt{(z+i)^2 + (2\lambda)^2}}{2\lambda} \cdot \frac{-z + i + \sqrt{(-z+i)^2 + (2\lambda)^2}}{2\lambda}$$

$$= \left( \frac{z+i}{2\lambda} + \sqrt{\left(\frac{z+i}{2\lambda}\right)^2 + 1} \right) \cdot \left( \frac{-z+i}{2\lambda} + \sqrt{\left(\frac{-z+i}{2\lambda}\right)^2 + 1} \right)$$

$$\le \exp\left(\frac{i}{\lambda}\right).$$

Hence, when $z \ge s$, we have $Q_{s,2\lambda}(z) \cdot Q_{s,2\lambda}(-z) < \exp\left(\frac{s(s+1)}{2\lambda}\right)$. Meanwhile, when $-s < z < s$, according to Lemma 8 in the appendix, we have:

$$Q_{s,2\lambda}(z) \le \exp\left(\frac{s(s+z)}{2\lambda}\right).$$

Therefore,

$$\Phi \le \exp\left(\frac{s(s+1)(\alpha-1)}{4\lambda}\right) \cdot \sum_{z=s}^{\infty} \left( \exp(\frac{s(\alpha-1)z}{2\lambda}) + \exp(-\frac{s(\alpha-1)z}{2\lambda}) \right) \Pr[Z = z]$$

$$+ \sum_{z=-s+1}^{s-1} \exp\left(\frac{s(\alpha-1)(s+z)}{2\lambda}\right) \Pr[Z = z]$$

$$\le \exp\left(\frac{s^2(\alpha-1)}{2\lambda}\right) \sum_{z=-\infty}^{\infty} \exp\left(\frac{s(\alpha-1)z}{2\lambda}\right) \Pr[Z = z]$$

$$= \exp\left(\frac{s^2(\alpha-1)}{2\lambda}\right) \mathbb{E}\left[ \exp\left(\frac{s(\alpha-1)Z}{2\lambda}\right) \right].$$

The moment generating function of $\mathrm{Sk}(\lambda, \lambda)$ is $\mathbb{E}[e^{tZ}] = e^{\lambda(e^t + e^{-t} - 2)}$. When $0 < t < 1$, we have $e^t + e^{-t} - 2 < 1.09t^2$. Thus, when $\lambda > \frac{s(\alpha-1)}{2}$,

$$\mathbb{E}\left[ \exp\left(\frac{s(\alpha-1)Z}{2\lambda}\right) \right] \le \exp\left(1.09\lambda(\frac{s(\alpha-1)}{2\lambda})^2\right) = \exp\left(\frac{1.09s^2(\alpha-1)^2}{4\lambda}\right).$$

As a result, when $\lambda > \frac{s(\alpha-1)}{2}$, we have $\Phi \le \exp\left(\frac{s^2(\alpha-1)}{2\lambda} + \frac{1.09s^2(\alpha-1)^2}{4\lambda}\right).$ □

---

**Algorithm 2:** Federated learning with distributed Skellam mechanism

---

**Input:** Private dataset $X = (x_1, \ldots, x_n)$. Sampling parameter $q$. $\mathcal{L}_2$ norm clipping bound $C > 0$.
  Quantization granularity $\gamma$. Conditional rounding parameter $k$. Noise parameter $\lambda$. Black-box
  secure aggregation protocol $\mathcal{A}(\cdot)$. Initial model parameters $\theta$. Number of iterations $T$.
**Output:** $\theta$ model parameters learnt on $X$.

1 **for** $h \in 1..T$ **do**
2 | The server shares the current model parameters $\theta$ to all clients.
3 | $\mathcal{C} \xleftarrow{u.a.r} \{1, 2, \ldots, n\}$ ;          // sample a subset of clients uniformly at random from all clients using
  | Poisson sampling with rate $q$.
4 | **for** $i \in \mathcal{C}$ **do**
5 | | $g_i \longleftarrow \nabla_\theta(x_i)$;          // gradient computation.
6 | | $g_i \longleftarrow Clip(g_i, C)$;          // gradient clipping based on $\mathcal{L}_2$ norm.
7 | | $g_i \longleftarrow g_i/\gamma$;          // quantization.
8 | | $g_i \longleftarrow Round(g_i, k)$;          // conditional rounding.
9 | $\tilde{g} \longleftarrow$ Distributed Skellam mechanism $(\{g_i\}_{i \in \mathcal{C}}, \lambda, \mathcal{A}(\cdot))$.
10 | $\tilde{g} \longleftarrow \gamma \cdot \tilde{g}$;          // re-scaling
11 | $\theta \longleftarrow Update(\theta, \tilde{g})$;          // model update based on the approximate gradient sum

---

Note that the privacy guarantee of DSM in Theorem 1 is independent of the data dimensionality $d$. Next we analyze the error incurred by Algorithm 1, which has the following bound:

**Corollary 1** (Error bound of the distributed Skellam mechanism). *With* $1 < \alpha < \frac{2n\lambda}{\Delta_\infty} + 1$, *the error incurred by Algorithm 1 that satisfies* $(\alpha, \tau)$*-RDP is bounded by*

$$Err_{\mathcal{M}} \leq \frac{1.09\alpha + 0.91}{2} \cdot \frac{d\Delta_2^2}{\tau}.$$

The proof for the above corollary is omitted for brevity, which directly follows from Theorem 1 and the linearity of expectation. Note that the error bound in the above corollary is comparable with (i.e., within a constant factor) that of adding continuous Gaussian noise in the centralized setting, which is $(\alpha - 1) \cdot d \cdot \Delta_2^2/(2\tau)$ (Mironov, 2017). Meanwhile, since Algorithm 1 satisfies RDP, by Lemmata 1 and 2, it allows the tight privacy accounting in DPSGD through composition and subsampling. Note that although the above analysis restricts the value of $\alpha$ to $1 < \alpha < 2n\lambda/\Delta_\infty + 1$, this constraint only affects the utility of DSM, not its privacy guarantees. Further, in practice, $n$ (i.e, the number of training data points) is usually much larger than $\Delta_\infty$, leading to a large range of values for $\alpha$. In addition, the optimal $\alpha$ (order of RDP) is often relatively small (e.g. less than 10) in our experiments, and the above range represents a sufficiently large space for tuning $\alpha$.

Comparing the theoretical results DSM with those of existing solutions described in Section 1, cpSGD (Agarwal et al., 2018) does not satisfy RDP, and does not have the corresponding composition and sub-sampling properties that are crucial for tight privacy analysis. Meanwhile, DDG (Kairouz et al., 2021) ensures $(\alpha, \tau')$-RDP with $\tau' = \frac{\alpha}{2} \cdot \min\left(\frac{1}{n} \cdot \frac{\Delta_2^2}{\sigma^2} + \frac{1}{2}\rho \cdot d, \left(\frac{1}{\sqrt{n}} \cdot \frac{\Delta_2}{\sigma} + \rho \cdot \sqrt{d}\right)^2\right)$ when each client independently injects discrete Gaussian noise of variance $\sigma^2$ to her data, where $\rho$ is defined as $\rho := 10 \cdot \sum_{k=1}^{n-1} \exp(-2\pi^2\sigma^2 \cdot \frac{k}{k+1})$. Note that the above $\tau'$ term increases with the data dimensionality $d$, which is not scalable for large $d$. In contrast, the privacy guarantee of DSM is independent of $d$. These advantages of DSM lead to significant utility gains in our target application: FL under differential privacy, described next.

### 3.3 FEDERATE LEARNING WITH DISTRIBUTED SKELLAM MECHANISM

We apply the proposed DSM described in Algorithm 1 to enforce DP on federated learning with distributed SGD, assuming that the clients have access to a black-box secure aggregation protocol. Algorithm 2 outlines the training process. In each iteration, the server releases the current model parameters to all clients (Line 2). Then, DSM randomly selects a subset of clients, whose identities are not known to the server (Line 3). Each client in the selected subset then computes the gradients based on the current model weights and her own data (Line 5), and performs gradient clipping (Line 6), which is a standard step introduced in DPSGD to bound the sensitivity of deep learning with

DP. After that, the client quantizes the gradients (Line 7), as required by typical MPC protocols for gradient aggregation. Here, a parameter $\gamma$ controls the quantization granularity: a smaller $\gamma$ leads to more precise quantization, and vice versa. Subsequently, the client randomly rounds the gradient to the integer grid $\mathbb{Z}^d$ (Line 8, explained soon). Next, we apply the distributed Skellam mechanism (Algorithm 1) to the integer-valued gradients (Line 9). Finally, the server re-scales the approximate gradient sum by a factor of $\gamma$ (Line 10) and updates the shared model accordingly (Line 11). We omit additional details on the updating process (e.g., learning rate schedule, weight decay, etc.) as they do not affect the general framework or the privacy guarantees. After repeating the above process for $T$ iterations, the training terminates, and the server obtains the final model weights $\theta$.

Next we clarify the conditional rounding step (Line 8 of Algorithm 2). This step is necessary since after gradient clipping (Line 6) and quantization (Line 7), we still have real-valued gradients, which are not compatible with the MPC protocol (i.e., the black-box secure aggregation protocol $\mathcal{A}$ used in DSM). To explain the conditional rounding in Line 8, we first define an element-wise unconditional rounding process as follows: for input $x \in \mathbb{R}$, $x$ is rounded to $floor(x)$ with probability $ceil(x) - x$, and to $ceil(x)$ otherwise. The problem with this unconditional gradient rounding process is that it introduces an additional $O(d)$ term on the $\mathcal{L}_2$ sensitivity of the resulting integer gradients. To mitigate this issue, Kairouz et al. (2021) propose to repeatedly run the unconditional rounding algorithm until the $\mathcal{L}_2$ norm of the rounded result satisfies a pre-defined condition. The stopping condition in (Kairouz et al., 2021) is rather complex, and increases the sensitivity of the resulting gradients by a factor of $O(\sqrt{d})$. Algorithm 2 adopts this general idea, but sets a much simpler stopping condition than the one in DDG, as follows: the rounded gradient's $\mathcal{L}_2$ norm must not exceed a pre-defined constant $k$ (e.g., 5 in our experiments) times $C/\gamma$, where $C$ is the $\mathcal{L}_2$ clipping bound in Line 6, and $\gamma$ is the quantization granularity in Line 7. With this stopping condition, the increase in $\mathcal{L}_2$ sensitivity in Line 8 is a constant independent of the gradient dimensionality $d$.

Regarding privacy guarantees of Algorithm 2, observe that each iteration of the training process can be seen as running the distributed Skellam mechanism on a random subset of rounded gradients. This is because the model sharing (Line 2), gradient sum re-scaling (Line 10), and model updating (Line 11) do not incur any additional privacy loss, as the updated model can be reconstructed by the re-scaled perturbed gradient sum, which, in turn, can be computed from the perturbed gradient sum. In addition, since the identities of the random subset of clients are not known to the server, the privacy guarantee benefits from amplification by subsampling. (We refer the reader to (Kairouz et al., 2021) for a detailed discussion on this issue.) Hence, the privacy guarantee of Algorithm 2 follows by applying the composition lemma (Lemma 1) and the amplification lemma (Lemma 2) on the privacy analysis of DSM (Theorem 1). A formal statement of the privacy guarantees of Algorithm 2 is presented in Theorem 2 in the appendix.

## 4    EXPERIMENTS

We evaluate the performance of the proposed solution DSM (Algorithm 2) on two classic benchmark datasets: MNIST (Lecun et al., 1998) and Fashion MNIST (Xiao et al., 2017), which contains grayscale images of handwritten digits and clothing, respectively. Both datasets represent 10-class classification tasks with $60,000$ training data records. We regard each data record in the training data as a client. For both tasks, we train a three-layer neural network with fully connected layers and ReLu activation, following previous work (Agarwal et al., 2018). We set the number of neurons per layers to 80, resulting in a model with $d = 63,610$ weights. For DDG and DSM, we use the same $\mathcal{L}_2$ clipping norm ($C = 1$) and quantization granularity ($\gamma = 0.1$). In addition, for cpSGD, we apply the tighter privacy analysis presented in (Koskela et al., 2021), which leads to improved prediction accuracy. For all experiments, we use the Adam optimizer (Kingma & Ba, 2015) with learning rate $\eta = 0.005$. We do not tune the hyper-parameters in favor of any particular solution and omit additional experiments on hyper parameter tuning. We remark that our approach is compatible with existing differentially private parameter tuning techniques (Gupta et al., 2010; Liu & Talwar, 2019), which is an orthogonal topic to this paper.

Our evaluation uses the $(\epsilon, \delta)$-DP (Definition 1) definition instead of RDP (Definition 3, since a competitor cpSGD supports the former but not the latter. We fix $\delta$ to $10^{-5}$, and vary the privacy parameter $\epsilon$ from $\{1, 2, 3, 4, 5\}$, batch size $m$ from $\{120, 240, 480, 960\}$. The model is trained for 1 epoch, i.e., when $m$ equals to $120, 240, 480,$ and $960$, we train the model for $500, 250, 125,$ and

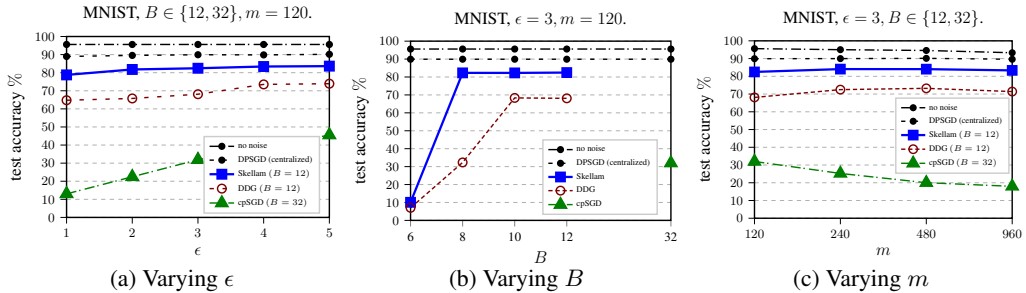

Figure 1: Evaluations on MNIST with varying privacy parameter $\epsilon$, bit limit per weight $B$, and batch size $m$.

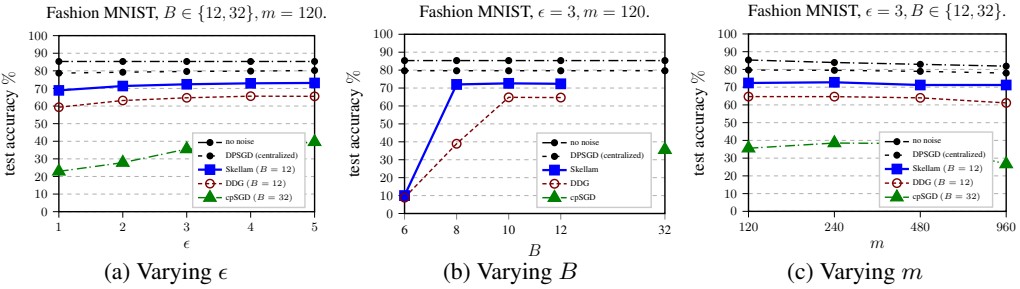

Figure 2: Evaluations on Fashion MNIST with varying privacy parameter $\epsilon$, bit limit per weight $B$, and batch size $m$.

62 rounds, respectively. For (DDG) and our solution DSM, we also vary the bit limit per weight $B$ from $\{6, 8, 10, 12\}$ bits; for cpSGD, we fix $B$ to 32. This is because the amount of noise injected by cpSGD is rather large, and smaller values of $B$ lead to very poor model utility. Note that a large $B$ leads to higher communication costs between the clients. We report the average test accuracy over 5 runs. The results are shown in Figures 1 and 2. In addition, we have also included the non-private baseline and the centralized-DP baseline DPSGD (Abadi et al., 2016b) in the figures.

Overall, the proposed solution DSM (marked as "Skellam" in the figures) significantly outperforms its competitors cpSGD and DDG on all settings. In particular, the accuracy improvement over DDG is consistently around 15% and 10% for MNIST and Fashion MNIST, respectively. From Figures 1(a) and 2(a), we observe that the performance gap expands as $\epsilon$ decreases, which corresponds to stronger privacy protection and requires higher noise levels. According to Figures 1(b) and 2(b), our solution is the only mechanism that achieves acceptable accuracy when the number of bits $B$ is as low as 8 per weight, which corresponds to a stringent constraint on the communication costs in the underlying MPC protocol. The test accuracy obtained with cpSGD is rather low (no more than 30%), even though it is given a generous value of $B = 32$. Finally, according to Figures 1(c) and 2(c), the accuracy for all methods remains stable with varying batch size $m$.

## 5   CONCLUSION

In this work, we propose the distributed Skellam mechanism (DSM), a novel solution for enforcing differential privacy on machine learning models built through an MPC-based federated learning process using distributed stochastic gradient descent. Compared to existing solutions, DSM achieves privacy guarantee that is independent of the dimensionality of the weight vector, while allowing tight privacy accounting due to its nice composition and sub-sampling properties. We conduct extensive experiments on two classic benchmark datasets and various practical settings, and the result demonstrate the consistent and significant accuracy gains DSM over existing solutions.

Regarding future work, we plan to further reduce the constant factor in the privacy analysis for DSM to improve model utility under the same level of privacy protection. Another promising direction is to open up the black box of the MPC protocol and perform careful privacy analysis with considerations for the details of MPC protocol, which might help lower the noise level further, leading to a more favorable privacy-utility trade-off for federated learning.

## REPRODUCIBILITY STATEMENT

We provide our source code for our distributed Skellam mechanism implementation for reviewing purposes, available at `https://anonymous.4open.science/r/Distributed_Skellam_Mechanism`. Our implementation simulates federated learning with a single machine.

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

# A APPENDIX

## A.1 PROOFS

**Lemma 4.** *Let $Q_{s,t}(v) = \prod_{i=1}^{s} \frac{v+i+\sqrt{(v+i)^2+t^2}}{t}$. Then, $\frac{I_{|v|}(t)}{I_{|v+s|}(t)} \le Q_{s,t}(v)$.*

*Proof.* By properties of the modified Bessel function of the first kind (Andrea & Pierpaolo, 2010), we have inequality

$$\frac{I_{v-1}(t)}{I_v(t)} < \frac{v + \sqrt{v^2+t^2}}{t}, \qquad v \ge 0$$

and recurrence relation

$$I_{v+1}(t) = I_{v-1}(t) - \frac{2v}{t} I_v(t)$$

which leads to

$$\frac{I_{v+1}(t)}{I_v(t)} < \frac{-v + \sqrt{v^2+t^2}}{t}, \qquad v \ge 0.$$

When $v < 0$, substituting $v$ for $-v-1$ in the previous formula,

$$\frac{I_{|v|}(t)}{I_{|v+1|}(t)} = \frac{I_{-v}(t)}{I_{-v-1}(t)} < \frac{v+1+\sqrt{(v+1)^2+t^2}}{t}$$

When $v \ge 0$, we also have

$$\frac{I_{|v|}(t)}{I_{|v+1|}(t)} = \frac{I_v(t)}{I_{v+1}(t)} < \frac{v+1+\sqrt{(v+1)^2+t^2}}{t}$$

Therefore, for every $v$,

$$\frac{I_{|v|}(t)}{I_{|v+s|}(t)} < \prod_{i=1}^{s} \frac{v+i+\sqrt{(v+i)^2+t^2}}{t} := Q_{s,t}(v).$$

$\square$

**Lemma 5.** *For any $w \ge 0$,*

$$w + \sqrt{w^2+1} \le e^w.$$

*Proof.* The inequality holds when $w = 0$. By taking the derivative, it can be verified that the function $e^w - (w + \sqrt{w^2+1})$ is increasing with respect to $w$. $\square$

**Lemma 6.** *For any $0 < a \le w$,*

$$\frac{w + \sqrt{w^2+1}}{(a-w) + \sqrt{(a-w)^2+1}} \le e^{2w-a}.$$

*Proof.* By Lemma 5, $w + \sqrt{w^2+1} \le e^w$. Then, by Lemma 5 again,

$$\frac{1}{(a-w) + \sqrt{(a-w)^2+1}} = (w-a) + \sqrt{(w-a)^2+1} \le e^{w-a}.$$

$\square$

**Lemma 7.** *For any $0 < a \le w$,*

$$(w + \sqrt{w^2+1}) \cdot ((a-w) + \sqrt{(a-w)^2+1}) \le e^a.$$

*Proof.* By Lemma 5, the inequality holds when $w = a$. By taking derivative, it can be verified that the left hand side is decreasing with respect to $w$. $\square$

**Lemma 8.** *When $-s < z < s$, we have*

$$Q_{s,2\lambda}(z) \leq \exp\left(\frac{s(s+z)}{2\lambda}\right).$$

*Proof.* Use Lemma 5. When $0 \leq z < s$,

$$Q_{s,2\lambda}(z) \leq \prod_{i=1}^{s} \exp\left(\frac{z+i}{2\lambda}\right) = \exp\left(\frac{s(s+1+2z)}{4\lambda}\right) \leq \exp\left(\frac{s(s+z)}{2\lambda}\right).$$

When $-\frac{s}{2} \leq z < 0$, using the equality

$$\frac{b+\sqrt{b^2+(2\lambda)^2}}{2\lambda} \cdot \frac{-b+\sqrt{(-b)^2+(2\lambda)^2}}{2\lambda} = 1,$$

we have

$$\begin{aligned}
Q_{s,2\lambda}(z) &= \prod_{i=1}^{s} \frac{z+i+\sqrt{(z+i)^2+(2\lambda)^2}}{2\lambda}\\
&= \prod_{i=-2z}^{s} \frac{z+i+\sqrt{(z+i)^2+(2\lambda)^2}}{2\lambda}\\
&\leq \prod_{i=-2z}^{s} \exp\left(\frac{z+i}{2\lambda}\right)\\
&= \exp\left(\frac{s(s+2z-1)}{4\lambda}\right)\\
&\leq \exp\left(\frac{s(s+z)}{2\lambda}\right).
\end{aligned}$$

When $-s < z < -\frac{s}{2}$, we have

$$\begin{aligned}
Q_{s,2\lambda}(z) &= \prod_{i=1}^{-2z-s-1} \frac{z+i+\sqrt{(z+i)^2+(2\lambda)^2}}{2\lambda}\\
&\leq 1\\
&\leq \exp\left(\frac{s(s+z)}{2\lambda}\right).
\end{aligned}$$

$\square$

## A.2 PRIVACY GUARANTEE OF ALGORITHM 2

**Theorem 2** (Privacy guarantee of Algorithm 2). *Let $\alpha \in \mathbb{Z}, 2 < \alpha < \frac{2n\lambda\gamma}{Ck} + 1$. Then for $\mathcal{L}_2$ norm clipping bound $C$, noise parameter $\lambda$, quantization granularity $\gamma$, conditional rounding parameter $k$, sampling parameter $q$, and iterations $T$, Algorithm 2 satisfies $(\alpha, \tau)$-Rényi Differential Privacy with*

$$\tau = T \cdot \frac{1}{\alpha-1} \log\left((1-q)^{\alpha-1}(\alpha q - q - 1) + \sum_{l=2}^{\alpha} \binom{\alpha}{l}(1-q)^{\alpha-l} q^l e^{(l-1)\tau(l)}\right), \tag{13}$$

*where $\tau_l$ is defined as $\tau_l := \frac{1.09l+0.91}{2} \cdot \frac{C^2 k^2}{2n\lambda\gamma^2}$, for $l = 2, \ldots, \alpha$.*

We omit the proof for brevity, which follows from our main theoretical result Theorem 1, as well as subsampling (Lemma 2) and composition (Lemma 1).

## A.3 OVERFLOWING IN MODULAR ARITHMETIC

The key to avoid overflowing is to restrain the perturbed gradient weights within the range of the finite integer filed, namely $[-2^{B-1}, 2^{B-1} - 1]$. We first review the concentration property of the symmetric Skellam distribution $Z \sim Sk(\lambda, \lambda)$ of mean $0$. Note that $Z$ has MGF $\exp(\lambda(\exp(\lambda) + \exp(-\lambda) - 2))$. In addition, since $x \cdot (\exp(x) + \exp(-x) - 2) \leq 1.09x^2$ for $0 \leq x < 1$. We have that $Z$ is sub-exponential with parameters $(\sqrt{2.18}, 1)$. Namely,

$$\mathbb{E}[\exp(tZ)] \leq \exp(1.09t^2), \text{for } |t| \leq 1. \tag{14}$$

From here we obtain the tail bound for $Z$,

$$\Pr[Z \geq x] \leq \exp(-x/2), \text{for } x \geq 2.18. \tag{15}$$

Hence, to avoid overflowing due to the Skellam noise, it suffices to set $2^B \geq h \log d$, where $d$ is the dimension of the gradient weights, and $h$ is some constant to balance the communication cost and the error due to overflowing. A relatively large $h$ limits the error due to overflowing while incurring a large communication cost; and vice versa. Note that a more precise reasoning can be obtained by observing that $Z$ approaches a normal distribution with mean $0$ and variance $2\lambda$, when $Z$ is of order $\sqrt{2\lambda}$. In fact, in our experiments, we observe that overflows are rare under a reasonable communication constraint. To be more specific, no overflow was observed when the number of bits per dimension reaches 10 or above, and very few overflows occured when the number of bits per dimension is 8 (once every a thousand weights).

## A.4 CONCURRENT WORK

Agarwal et al. (2021) also suggest using the Skellam noise in the distributed DP setting. Different from our work, their privacy analysis for the Skellam noise is dependent on both the L2 norm and L1 norm of the private input vector. Since the L1 norm of a high dimensional vector can be much larger than its L2 norm, the privacy analysis we present is tighter. Further tightening the analysis is a promising future direction.

