# OpenReview forum: "Distributed Skellam Mechanism: a Novel Approach to Federated Learning with Differential Privacy"
_ICLR.cc/2022/Conference — ICLR 2022 Submitted_

### Official Review · Reviewer_QVkQ · 2021-11-01

**Correctness:** 4
**Technical Novelty And Significance:** 3
**Empirical Novelty And Significance:** Not applicable
**Recommendation:** 8
**Confidence:** 4

**Main Review:**

Strong points:
* The proposed method addresses some main weaknesses in the existing methods (mainly: avoid bad scaling with dimensionality that cripples the discrete Gaussian mechanism, allow for RDP-based privacy accounting)
* The results seem to work clearly better than existing methods

Weak points:
* There is no discussion on how tight the resulting privacy bounds are, i.e., how much do you lose when using the inequalities in Section 3.2
* I am not totally sure if the comparison against existing work is done entirely properly (see questions for authors below for details)

Questions and comments for the authors:
1) Can you quantify how tight or loose the RDP bound in Thm1 actually is?

2) On p. 2 you state that "existing theoretical tools for analyzing binomial noise aggregation leads to rather loose bounds". I get the impression that you mean RDP bounds. Still, there are other accounting methods like the Fourier accountant that operates directly in with (epsilon,delta)-DP, and Gaussian DP (GDP) based accounting. For example, looking at [1], there does seem to be calculations for some tighter bounds for the binomial mechanism. Is there some reason not to consider this in the comparisons?

3) Very minor comment: there seem to be lot of small typos, please fix.

References:
[1] Koskela et al. 2021: Tight Differential Privacy for Discrete-Valued Mechanisms and for the Subsampled Gaussian Mechanism Using FFT.

**Summary Of The Paper:**

### update after the discussions

I am satisfied with the responses from the authors' and the updated paper. I therefore recommend accepting the paper.

---

The paper considers the problem of distributed differentially private (DP) learning using black-box secure multi-party computation (MPC) for aggregating gradients for learning the model.

In principle, using e.g. secure aggregation allows each party to add a small amount of noise scaled so that the noise level after aggregation matches what one needs with a trusted central party. However, since the existing tools (mainly Gaussian mechanism) typically assume continuous space, while MPC works with discrete values, this does not work in practice: the main problem is that the discretised noise is not infinitely divisible so the sum is not guaranteed to follow the same distribution as the individual contributions.

To remedy the problem, the authors propose the distributed Skellam mechanism, which is both dicrete and infinitely divisible. The authors show that the Skellam mechanism privacy cost can be calculated using Rényi DP (RDP), and continue to show that it performs significantly better than the existing methods based on binomial noise and discrete Gaussian noise using MNIST and Fashion MNIST data for testing.

**Summary Of The Review:**

Overall I think this is a nice paper. However, I would like to hear the author response (questions 1 and 2) before recommending acceptance.

---

> ### Author Response · Authors · 2021-11-15
> **Response to Reviewer QVkQ**
>
> __Comment 1.__ _Can you quantify how tight or loose the RDP bound in Thm1 actually is?_
>
> __Response.__ As described in the paragraph below Corollary 1 in the revised paper, our bound is similar to that of the continuous Gaussian mechanism, which applies to the centralized setting that does not have the restriction that gradient values must be represented as finite field elements. In particular, observe that Corollary 1 has a term $(1.09\alpha + 0.91)$; if we substitute this term with simply $\alpha$, we would arrive at the error bound of the continuous Gaussian mechanism [Mironov 2017]. For a reasonably large $\alpha$, $(1.09\alpha + 0.91)$ is not much larger than $\alpha$, indicating that our bound is tight.
>
> __Comment 2.__ _On p. 2 you state that "existing theoretical tools for analyzing binomial noise aggregation leads to rather loose bounds". I get the impression that you mean RDP bounds. Still, there are other accounting methods like the Fourier accountant that operates directly in with (epsilon,delta)-DP, and Gaussian DP (GDP) based accounting. For example, looking at [1], there does seem to be calculations for some tighter bounds for the binomial mechanism. Is there some reason not to consider this in the comparisons?_
>
> __Response.__ Good point. Our current implementation of the binomial mechanism strictly follows that in their paper, which involves only one epoch with disjoint batches, and, thus, does not require composition at all. Specifically, before the training starts, we randomly partition the 60,000 training data points to a number of disjoint subsets, e.g., 500 subsets each containing 120 data points. In each epoch, the model is updated based on one subset only. To ensure fair comparison, in our experiments, we also limit the proposed solution DSM to one epoch only. Hence, the performance advantage of DSM over the binomial mechanism is not due to the composition properties of the former.
>
> Theoretically, it is possible to run the binomial mechanism for multiple epochs, by using the same batches in every epoch (i.e., no shuffling after an epoch); then, on each batch, the mechanism is applied multiple times, and privacy loss can be accounted by a composition theorem such as the Fourier accountant. The design and implementation of this method (which is beyond the original paper on the binomial mechanism) is left as future work due to the time constraints in the rebuttal phase. On the other hand, considering that the performance gap between DSM and the binomial mechanism is rather large for one epoch, their relative performance is unlikely to change when we run multiple epochs, especially considering that DSM supports both composition and subsampling, and has tight bounds in its privacy analysis.
>
> __Comment 3.__ _Very minor comment: there seem to be lot of small typos, please fix._
>
> __Response.__ Revised as suggested.

---

### Official Review · Reviewer_85EN · 2021-11-02

**Correctness:** 4
**Technical Novelty And Significance:** 2
**Empirical Novelty And Significance:** 2
**Recommendation:** 5
**Confidence:** 4

**Main Review:**

Strengths:
1. The paper is clearly structured and the theoretical proof is very logical.
2. Some nice properties of Skellam mechanism are considered, i.e., sum of Skellam random variables is Skellam distributed.


Weaknesses:
1. Lack of novelty. Similar to this paper, the idea of applying the Skellam mechanism to federated learning has already been explored, and extended to high-dimensional settings in a recent paper. https://arxiv.org/abs/2110.04995
2. Doubt about the accuracy-privacy trade-offs. Due to composition theorems applicable in learning settings, the mechanism will be applied many times.

**Summary Of The Paper:**

This paper presents a mechanism based on Skellam distribution (called Distributed Skellam Mechanism (DSM)) to prevent privacy leakage for federated learning. It provides analysis of privacy guarantee in the decentralized setting. Specifically, DSM is shown to be both RDP and (\epsilon, \delta)-DP. Also, DSM is applied to differentially private federated learning with distributed SGD and quantized gradients.

**Summary Of The Review:**

Compared to other approaches to address privacy issues in joint learning, the Skellam mechanism is very compatible with MPC. However, a similar work has already been done.

---

> ### Author Response · Authors · 2021-11-15
> **Response to Reviewer 85EN**
>
> __Comment 1.__ _Lack of novelty. Similar to this paper, the idea of applying the Skellam mechanism to federated learning has already been explored, and extended to high-dimensional settings in a recent paper. https://arxiv.org/abs/2110.04995_
>
> __Response.__ This recent paper was released on arXiv on October 11th, 2021, which is 5 days after our paper was submitted to ICLR 2022 and released on OpenReview. According to the ICLR 2022 reviewer guide (https://iclr.cc/Conferences/2022/ReviewerGuide, FAQ), this is considered a contemporaneous paper, whose existence does not undermine the novelty of our paper. On the contrary, the fact that a contemporaneous paper (with similar ideas as ours) got accepted at NeurIPS’21 actually shows these ideas are deemed novel, at least by the NeurIPS reviewers.
>
> In addition, the results and proof techniques in the arXiv paper are different from ours. Specifically, the arXiv paper’s privacy bound for the Skellam noise requires upper-bounding both the L1 and L2 sensitivity of the input, whereas our result is based only on the L2 sensitivity. As a consequence, our result leads to a tighter privacy bound when the input’s L1 sensitivity is considerably larger than its L2 sensitivity.
>
> __Comment 2.__ _Doubt about the accuracy-privacy trade-offs. Due to composition theorems applicable in learning settings, the mechanism will be applied many times._
>
> __Response.__ Indeed, the mechanism needs to be applied many times -- once for every iteration of the SGD algorithm. To alleviate the problem, we adopt the standard trick of subsampling (Lemma 2), which leads to tight analysis on the privacy loss for the iterative SGD process. The main idea is that although the mechanism is applied many times, each time it is only applied on a random sample of the training set. Note that in an SGD iteration, when a data record is not selected by the sampling, its privacy cost is 0; meanwhile, when the size of the sample set (i.e., batch size)  is small, the probability that a record is selected is small. A privacy accountant module (first proposed in DP-SGD by Abadi et al. 2016) keeps track of the total privacy loss in each epoch, and we determine the total number of epochs accordingly based on the privacy budget. In fact, supporting subsampling in the proposed distributed Skellam mechanism is highly non-trivial, and is one of the major contributions of the paper.

---

> > ### Comment · Reviewer_85EN · 2021-11-29
> > **Response**
> >
> > Thank you for the responses, which addressed some of my concerns. I am still not comfortable of accepting a paper which is very similar to another recently accepted paper to NeurIPS. I will keep my score and let the area chair and program chair decide on it.

---

> > > ### Author Response · Authors · 2021-12-01
> > > **Response**
> > >
> > > Putting aside ICLR rules, we understand your concern on the similarity of this paper and the arXiv one accepted at NeurIPS’21. One more thing we would like to point out is that these two papers present similar algorithms, but very different analysis and proof techniques. In a nutshell, the analysis in the NeurIPS’21 paper, while highly nontrivial, follows a conventional flow of thoughts in the differential privacy literature, which led to a privacy guarantee that relies on upper-bounding both the L1 and L2 sensitivity of the input. In contrast, our analysis applies a novel proof technique that eliminates the L1 norm term in our privacy guarantee.
> > >
> > > Specifically, a key step in both papers is to analyze the Renyi divergence between two shifted symmetric Skellam distributions, defined as the expected ratio between two modified Bessel functions of the First Kind on a Skellam distribution.
> > >
> > > * The NeurIPS’21 paper follows the standard trick (e.g., also used in the analysis of previous work DDG) of applying the triangle inequality of Renyi divergence, which leads to the L1 norm term in their final privacy guarantee.
> > > * To avoid the L1 norm term, this paper (starting from the paragraph after Inequality 12 on Page 6) does not use known properties of Renyi divergence, and instead attacks the problem directly using basic mathematical tools. This leads to rather long and heavy formulae at the beginning, and yet within a few steps, most terms get canceled out, resulting in a clean bound.
> > >
> > > Our proof technique can be a new addition to the toolbox of differential privacy mechanism designers. Meanwhile, the clean bound without the L1 norm term may significantly simplify the design of protocols and mechanisms built on top of DSM, which is a direction we have been working on.

---

### Official Review · Reviewer_VptB · 2021-11-02

**Correctness:** 3
**Technical Novelty And Significance:** 3
**Empirical Novelty And Significance:** 3
**Recommendation:** 6
**Confidence:** 4

**Main Review:**

Pros:

1. The privacy guarantee of DSM is non-trivial yet the analysis is clean.


2. From Theorem 1, the variance of the DSM that preserves an $(\alpha, \tau)$-RDP is very close to the variance of the (centralized) Gaussian mechanism. This thus results in a small utility loss compared to the centralized model.


3. The experimental results also support that DSM outperforms the existing DDG scheme in some parameter regimes, such as when the communication budget is limited.


Cons:

1. Theorem 1 only holds when $ 1 < \alpha < \frac{2n\lambda}{\Delta_\infty}+1$. Is this the particularly interesting parameter regime in practice? Since the proposed algorithm does not perform random rotation on $x$ in the pre-processing steps, in the worst-case scenario, $\Delta_\infty$ can be as large as $\Delta_2$. I guess it should be fine if $n\lambda$ is large enough, but it would be great if the authors can briefly comment on whether $ 1 < \alpha < \frac{2n\lambda}{\Delta_\infty}+1$ is a reasonable regime or not.


2. In Algorithm 1 and 2, it seems that the summation is over $\mathbb{Z}$ instead of over $\mathbb{Z}_m$. In other words, there should be a modular clipping step before the aggregation (because SecAgg only works on a finite group). Moreover, similar as in [KLS 2021], when aggregating with a coordinate-wise modular sum, there should be a random rotation step that spreads the mass of $x_i \in \mathbb{R}^d$ evenly to each coordinate. I believe all these issues can be fixed by simply pre-processing $x_i$ with a random rotation and replacing the summation with modular sum though.


3. Following the previous point, since the aggregation should be carried out under modular arithmetic, Corollary 1 may not hold since the error due to modular clipping should also be incorporated.


4. In the experiments, DSM is compared with DDG and cpSGD. It would be good if the authors can also include the accuracy of the (centralized) Gaussian mechanism as a baseline, which would let us know how close the DSM is to the centralized error.

**Summary Of The Paper:**

This paper studies federated learning under the distributed DP framework [KLS 2021] and proposes the distributed Skellam mechanism (DSM). Compared to the existing approach [KLS 2021] that uses distributed discrete Gaussian (DDG) noise, DSM perturbs each local gradient with independent Skellam noise. This gives the advantage that the privacy guarantee is independent of the dimensionality of the gradients; further, DSM allows tight privacy accounting due to the nice composition and sub-sampling properties of the Skellam distribution and hence enjoys a better constant compared to the DG. Experimental results also imply that the DSM improves the previous DDG scheme (proposed in [KLS 2021]) when the communication is limited, say 12 bits per parameter.

**Summary Of The Review:**

Overall, the paper is well-written and the proofs are easy to follow. The contribution is solid since the result directly improves the previous DDG scheme. However, there are some minor issues that the authors should clarify or address.

---

> ### Author Response · Authors · 2021-11-15
> **Response to Reviewer VptB**
>
> __Comment 1.__ _Theorem 1 only holds when 1<α<2nλΔ∞+1. ...but it would be great if the authors can briefly comment on whether 1<α<2nλΔ∞+1 is a reasonable regime or not._
>
> __Response.__ Good question. First of all, note that restricting the value of $\alpha$  to a specific range does not affect the privacy guarantees of the proposed method, only its utility, i.e., if the proposed method can be shown to guarantee differential privacy with a value of $\alpha$ outside this range, then we have a larger space for hyperparameter tuning, which might lead to better accuracy performance. In our experiments, through manual investigations, we found that the optimal value of $\alpha$ is usually small. Hence, when $n$ is reasonably large, this parameter regime often leads to good performance. Rigorous analysis on how our range of $\alpha$ affects performance is an interesting direction for future research.
>
> __Comment 2.__ _In Algorithm 1 and 2, it seems that the summation is over Z instead of over Zm. In other words, there should be a modular clipping step before the aggregation (because SecAgg only works on a finite group)._
>
> __Response.__ We have revised the paper (on Page 5) to clarify that the proposed solution DSM does perform modular clipping, i.e., the summations are indeed over $\mathbb{Z}_m$.
>
> __Comment 3.__ _Moreover, similar as in [KLS 2021], when aggregating with a coordinate-wise modular sum, there should be a random rotation step that spreads the mass of xi∈Rd evenly to each coordinate. I believe all these issues can be fixed by simply pre-processing xi with a random rotation and replacing the summation with modular sum though._
>
> __Response.__ The random rotation step in [KLS 2021] does not affect privacy, only utility (i.e., by reducing the probability of overflow). While this pre-processing step works well in their method according to [KLS 2021], we choose not to apply it to the proposed solution DSM, since (i) through manual investigations of the experimental results, we found that even without random rotation, overflowing does not happen often, even with a very small number of bits (e.g., 8) per dimension in the integer representation; with 10-12 bits per dimension, overflows are very rare, and (ii) a selling point of DSM is that its utility is independent of the data dimensionality $d$ ; however, the random rotation step would lead to a sensitivity that depends on $d$, which adversely affects utility performance as we found through experiments.
>
> __Comment 4.__ _Following the previous point, since the aggregation should be carried out under modular arithmetic, Corollary 1 may not hold since the error due to modular clipping should also be incorporated._
>
> __Response.__ It is true that the analysis in Corollary 1 is limited to the case that no overflow happens. In the revised paper, we have added Appendix A.3, which extends the analysis to capture the error due to modular clipping. To reduce the probability of overflow, we also adopt the practice in [KLS 2021] that selects the parameters such that the variance due to DP noise is much smaller than $2^B$.
>
> __Comment 5.__ _In the experiments, DSM is compared with DDG and cpSGD. It would be good if the authors can also include the accuracy of the (centralized) Gaussian mechanism as a baseline, which would let us know how close the DSM is to the centralized error._
>
> __Response.__ Revised as suggested.

---

> > ### Comment · Reviewer_VptB · 2021-11-20
> > **Additional comments**
> >
> > Thanks for the clarification, I am happy with the response, and my view basically remains the same.
> >
> > One additional comment regarding your response on random rotation: instead of doing random rotation to obtain a l_infty bound, which gives an additional $\log d$ term, a more sophisticated preprocessing method is to compute Kashin's representation [Kashin 1977, Lyubarskii 2011] (see the mean estimation scheme in [Chen 2020] for example). This can save the additional $\log d $ factor for free and thus the bounded $\ell_infty$ assumption is not necessary.
> >
> > [Kashin 1977] Section of some finite-dimensional sets and classes of smooth functions
> > [Lyubarskii 2011]  Uncertainty principles and vector quantization
> > [Chen 2020] Breaking the communication-privacy-accuracy trilemma

---

> > > ### Author Response · Authors · 2021-11-20
> > > **Additional response**
> > >
> > > Thanks for the pointer. [Chen et al. 2020] focuses on the local differential privacy (LDP) setting, which is very different from our problem, i.e., distributed differential privacy with secure-multiparty computation (MPC). In particular, in [Chen et al. 2020], each individual user first converts her $d$-dimensional data using Kashin’s representation in a preprocessing step, which obtains a $c'd$-dimensional vector (where $c'$ is a constant). Then, the user quantizes each element of this vector to a single bit, resulting in a $c'd$-bit binary vector, and subsequently samples $b$ bits to form a sparse binary vector. In the LDP setting of [Chen et al. 2020], the user sends the $b$-bit representation of the sparse vector to the server, and the latter expands it to the original $c'd$-bit form via a shared random seed. However, in our setting, the set of all users need to collectively participate in an MPC protocol such as SecAgg, and it is a challenging problem to aggregate sparse representations of vectors under limited bandwidth. Although there is some recent work in the security community on this topic (e.g., Schoppmann et al. “Make Some ROOM for the Zeros: Data Sparsity in Secure Distributed Machine Learning”, CCS’19), to our knowledge, there does not yet exist a protocol for aggregating sparse vector representations with comparable security guarantees as the classic SecAgg protocol. Hence, the technique in [Chen et al. 2020] does not directly apply to our problem, and it is an interesting direction for future work.

---

### Official Review · Reviewer_ecND · 2021-11-07

**Correctness:** 2
**Technical Novelty And Significance:** 2
**Empirical Novelty And Significance:** 2
**Recommendation:** 3
**Confidence:** 4

**Main Review:**

Strengths of the paper:

They study a problem that is of fundamental importance in privacy preserving machine learning. The use of Skellam distribution seems novel and interesting.

Weaknesses of the paper:

Firstly from a theoretical point of view, the contribution seems quite incremental. In the abstract the authors claim "highly non-trivial" analysis -- which is nothing more than elementary calculus and statistical analysis. The central theoretical contribution seems to be a differentially private multi-party protocol to compute sum of vectors. In a nutshell, the protocol is simply to add noise (based on "Skellam distribution") by each participant and adding them up in a distributed fashion.

Secondly, I feel the experimental section is unsound and unconvincing. For example, getting an accuracy of ~80% on MNIST is unacceptable. To some perspective, the original paper on differentially private deep learning (by Abadi et al 2016) yields an accuracy of 95% on MNIST. Further, there is no "non-private" baseline provided.

**Summary Of The Paper:**

The main contribution of the paper is a differential private mechanism for federated learning. Further, they provide experimental evaluation of the same using MNIST and Fashion MNIST datasets.

**Summary Of The Review:**

There are significant drawbacks both from a theoretical and empirical point of view. I am not sure how to strengthen the paper theoretically, but I think from an empirical point of view a more thorough comparison with various DP mechanism (Abadi et al. for instance) and including a baseline would strengthen the paper.

---

> ### Author Response · Authors · 2021-11-15
> **Response to Reviewer ecND**
>
> __Comment 1.__  _Firstly from a theoretical point of view, the contribution seems quite incremental. ... In a nutshell, the protocol is simply to add noise (based on "Skellam distribution") by each participant and adding them up in a distributed  fashion._
>
> __Response.__ It is true that the proposed solution DSM is built upon nothing more than elementary calculus and statistical analysis. This, however, does not contradict with the fact that the construction of DSM is nontrivial -- elementary calculus and statistical analysis are just the right tools for the job. (As an analogy, Pfizer’s Covid19 pill Paxlovid is designed with nothing more than organic chemistry and biology, but that does not make the pill any less significant or valuable.) We also note that all our competitors, such as cpSGD and DDG, are based on elementary calculus and statistical analysis, and all these protocols simply add noise in a distributed fashion.
>
> __Comment 2.__  _Secondly, I feel the experimental section is unsound and unconvincing. For example, getting an accuracy of ~80% on MNIST is unacceptable. To some perspective, the original paper on differentially private deep learning (by Abadi et al 2016) yields an accuracy of 95% on MNIST._
>
> __Response.__ The original paper on differentially private deep learning (by Abadi et al 2016, referred to as DP-SGD) relies on the assumption that a trusted, centralized party has access to all private data. Our target problem is fundamentally different from that in Abadi et al 2016, since we operate in a decentralized setting involving multiple parties who do not trust each other, meaning that each party must keep its data and gradients strictly confidential from other participating parties. Hence, DP-SGD is not applicable to our setting, and is not directly comparable with the proposed method. Regarding the accuracy results, although around ~80% accuracy may not appear high, it represents an over 10 percentage points improvement compared to the previous SOTA for our problem setting, which is confirmed in the experiments under various parameter combinations.
>
> __Comment 3.__  _Further, there is no "non-private" baseline provided._
>
> __Response.__ As suggested, in the revised paper, we have included the results of both DPSGD and a non-private baseline in the experiments section.

---

### Decision · Program_Chairs · 2022-01-20

**Decision:**

Reject

**Comment:**

This paper proposes the distributed Skellam mechanism for differentially private federated learning that relies on secure aggregation. Since multi-party computation protocols rely on finite precision, the Skellam distribution meets the criteria of closure under addition and discreteness. During the discussion, the reviewers raised a concern that the proposed idea is highly similar to a recently published NeurIPS paper (https://arxiv.org/abs/2110.04995). However, since the timelines of the two papers are close, they can be considered concurrent work. Both results advance the study of private federated learning that leverages secure aggregation techniques. Through a more in-depth comparison, the current paper also provides sufficiently different proof techniques than the ones in the concurrent paper. The authors should provide an extensive comparison between their work and the NeurIPS paper in their next revision.

While the paper provides new results, there are several concerns in the reviews. First, even though the proof techniques are different from concurrent work, the reviewers still think that the main technique of Skellam perturbation has limited novelty. Second, the presented experiments also appeared to be quite weak. For example, the accuracy on MNIST is much lower than simple baselines in earlier work. While the authors tried to justify this reduction of accuracy through their decentralized training setting, the argument is not fully convincing. In particular, even though the noise addition is done in a decentralized fashion, the proposed algorithm is still subject to the same (standard) differential privacy constraint (as opposed to local differential privacy). The authors could consider improving the experiments or providing a more principled justification for the reduction of accuracy in their algorithm. Due to these issues, the paper does not clear the bar for acceptance at ICLR.